# CALIBRATED ON AVERAGE, BUT NOT WITHIN EACH SLICE: FEW-SHOT CALIBRATION FOR ALL SLICES OF A DISTRIBUTION

## ABSTRACT

Recent work has uncovered promising ways to extract *well-calibrated* confidence estimates from language models (LMs), in which the model's confidence accurately reflects the probability that the answer is correct. However, while a model may be well-calibrated on average over some input distribution, the same model can actually be significantly miscalibrated within any narrower slice of the full distribution. For example, we find that a model may be well-calibrated over multiple-choice exam questions, but this calibration is the result of systematic overconfidence in one subject (e.g. math) getting balanced out by systematic underconfidence in another subject (e.g. history). In practice, being calibrated within narrower slices of a distribution is important because the full distribution is often formed from the queries of individual users who each only care about a narrower slice. In this work, we propose a new framework for calibrating models on any given slice of a distribution, using just a few unlabeled samples from that slice. Specifically, we train a model that approximates the precision-threshold curve for any given slice by using its few-shot samples to predict the LM's empirical precision at various confidence thresholds. This allows us to directly identify slice-specific thresholds above which the LM's predictions can be trusted (e.g. for a target precision of 90), and below which it should abstain. We also show that the precision curve can be mapped back to the classic calibration curve, which can guide adjusting the LM confidence to achieve lower calibration error. Experiments show that our fewshot recalibrator consistently outperforms existing calibration methods, for instance improving calibration error for PaLM2-Large on MMLU by 16%, as compared to temperature scaling.

## 1 INTRODUCTION

Knowing when to trust a model's predictions is typically mapped to the concept of calibration where the model's confidence estimate for a prediction reflects how likely it is to be correct. Language models (LMs) have recently been shown to be well-calibrated in a number of settings (Kadavath et al., 2022; Xiao et al., 2022; Kuhn et al., 2023; OpenAI, 2023). However, we find that while they may be well-calibrated for broader distributions (e.g. mixtures of a number of domains), LMs can be significantly miscalibrated for narrower slices of that broad distribution (e.g. individual domains).

For instance, Figure 1 shows an LM that is well-calibrated on questions from the diverse combination of five domains—*abstract algebra*, *business ethics*, *virology*, *high school chemistry* and *global facts*. While the combined calibration curve on the left appears near-perfect with a low expected calibration error (ECE), curves for the individual domains appear significantly miscalibrated in comparison, with the least calibrated domain *virology* having a 250% higher error. This miscalibration problem is hidden for the combined distribution because overconfidence in some domains cancels out underconfidence in others. This illustrates a key problem: LMs are not well-calibrated for meaningful slices of broader distributions. This is particularly relevant in practice where users querying an LM rarely sample from a broad combination of distributions at any given time, and are more likely to sample from slices like *abstract algebra* or *virology*. Our goal is to recalibrate LMs for each of these fine-grained distributions, to allow users to reliably determine when predictions can be trusted.

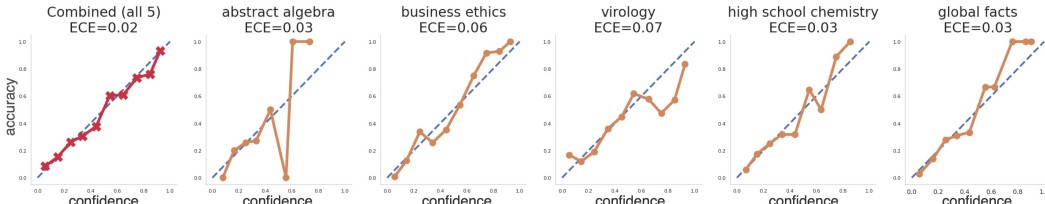

Figure 1: An example of the illusion of LM calibration. For a combination of five domains, the model is well-calibrated with a calibration error of 0.02 (the first plot). However, the same model is miscalibrated on the the five individual domains, each with a higher calibration error.[1]

In order to recalibrate a model on specific slices of a broader distribution, we propose fewshot recalibration—a new framework that uses a small number of unlabeled samples from the given slice to predict its precision curve. The precision curve maps a given confidence threshold to the corresponding precision for all examples with confidence higher than that threshold. We find that predicting precision curves is useful because they are flexible and can be used to achieve a diverse set of downstream goals, including recovering the more traditional calibration curves. We simulate slices for training and evaluation by starting with a broad distribution of queries, such as the five domains from Figure 1, and creating a large number of narrower distributions as a weighted mixture of a smaller number of domains, such as 80% *abstract algebra* and 20% *virology*. Then, we randomly sample a small number of queries from each slice and train a model to use this unlabeled fewshot sample to predict the corresponding precision curve. Note how this setup mimics the real-world setting where given a small set of a user's queries, our approach recalibrates the LM for that user's slice of the broader distribution.

We train our fewshot calibrator to recalibrate LLaMA-65B (Touvron et al., 2023) and PaLM2-Large (Anil et al., 2023) on the MMLU (Hendrycks et al., 2021) and XNLI (Conneau et al., 2018) datasets, which already categorize examples into domains allowing us to easily create slices. We evaluate our fewshot recalibrator against a variety of baselines in three settings: (1) achieving a desired level of target precision by identifying slice-specific confidence thresholds, (2) reducing calibration error per slice, and (3) maximizing utility by selecting the optimal slice-specific threshold, below which the model should abstain. Overall, we find that our fewshot calibrator consistently outperforms existing methods for calibration in all three settings. For PaLM2-Large on MMLU, our calibrator achieves a 21% higher success rate for achieving a target precision of 90 and a 16% lower calibration error on the test set slices, compared to directly using the precision and calibration curves for the combined distribution over all domains.

## 2 THE ILLUSION OF LM CALIBRATION

Calibration is a key tool for knowing when language model predictions can be trusted and when they should abstain or defer to experts. However, we find that even though LMs appear to be well-calibrated on average, they are significantly miscalibrated in finer-grained settings.

In this work, we study LM calibration for multiclass classification: let $x \sim p$ be the input drawn from the query distribution and $y \in \{1, \cdots, K\}$ be the output class. Let $p_{\text{LM}}(y \mid x)$ denote the model probability, which is also the model's confidence. Let $\hat{y} = \arg\max_y p_{\text{LM}}(y \mid x)$ be the model's prediction, and $y^*$ be the ground truth label.

### 2.1 MEASURING CALIBRATION

Calibration expresses how closely a model's confidence estimate for a prediction is aligned with the true probability that the prediction is correct, as measured by accuracy. We use $\text{acc}(\mathcal{B}) = \mathbb{E}_{(x,y^*,\hat{y}) \in \mathcal{B}} \mathbb{1}(\hat{y} = y^*)$ to denote the model's accuracy for the set $\mathcal{B}$, and $\text{conf}(\mathcal{B}) = \mathbb{E}_{(x,y^*,\hat{y}) \in \mathcal{B}} p_{\text{LM}}(\hat{y} \mid x)$ denotes the model's confidence on this set.

**Expected Calibration Error (ECE)** This is the canonical metric which measures $L_1$ distance between the confidence and accuracy (Naeini et al., 2015). To measure ECE, we first group all the

---

[1]Although a smaller sample size in MMLU can cause some jaggedness, our experiments on XNLI confirm this finding for larger sample sizes as well.

$N$ predictions into $M$ equally sized bins based on their confidence estiamtes, denoted as $B_1 \cdots B_M$. We then calculate the average confidence and accuracy of each bin, and compute the ECE of the LM under this query distribution $p(x)$:

$$\text{ECE}(p_{\text{LM}}, p) = \sum_{i=1}^{M} \frac{|B_i|}{N} |\text{conf}(B_i) - \text{acc}(B_i)|$$

Perfectly calibrated models have ECE $= 0$ i.e. model confidence matches expected accuracy at all confidence levels. For example, suppose there are 100 examples, each with confidence 0.8, we expect that 80 of the examples are correctly classified.

**Calibration Curves**  Also known as reliability diagrams, these curves are a visual representation of model calibration, plotting the expected model accuracy as a function of model confidence (De-Groot & Fienberg, 1983; Niculescu-Mizil & Caruana, 2005). Well-calibrated models lie close to the diagonal ($y = x$). Figure 1 shows example curves with respect to different query distributions $p(x)$.

## 2.2  POOR CALIBRATION ON SLICES OF DISTRIBUTIONS

We often study LM calibration for aggregate query distributions ($p$). But these are often composed of mixtures of meaningful finer-grained distributions: $p(x) = \sum_{d \in \mathcal{D}} \alpha_d p_d(x)$, where $\mathcal{D}$ denotes a set of domains, and each $p_d$ denotes the input distribution of domain $d$, with relative frequency $\alpha_d$. For instance, OpenAI (2023) and Kadavath et al. (2022) have reported LM calibration on MMLU, which consists of 57 individual domains like *abstract algebra*, *high school chemistry* etc. However, in practice, users querying an LM at a given point rarely sample from a broad aggregate distribution. They are more likely to sample from meaningful slices, like queries from *abstract algebra* alone.

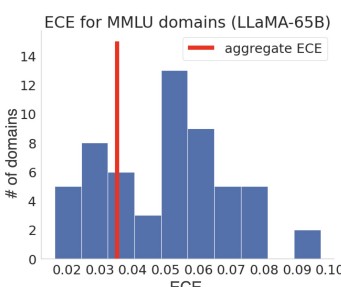

Figure 2:  A histogram of ECE scores for LLaMA-65B on 57 MMLU domains. The red line shows ECE for all the domains combined. We can see the aggregate ECE is lower than most domains, hiding the underlying miscalibration problem.

To better understand the reliability of model predictions in both settings, we measure calibration of LLaMA-65B on combined MMLU ($p$), similarly to previous work, and also measure calibration on each domain separately. As expected, the model is well-calibrated on $p$. However, we find that the LM is significantly miscalibrated for most domains. This is shown in (Figure 2) where the aggregate ECE is lower than that of most domains. It appears that the miscalibration problem is hidden for the broader distribution because overconfidence in some domains cancels out underconfidence in others. Figure 1 shows a qualitative example to illustrate this using five domains from MMLU: while the model is well-calibrated for the combined distribution with a curve that falls nearly on the diagonal and has the lowest ECE, it appears to be significantly miscalibrated for each of the five domains. The worst calibration error for *virology* is about $250\%$ higher than that of the combined distribution. These results show that LMs are not always well-calibrated for meaningful slices of broader distributions, making it hard for users to know when they can trust model predictions.

## 3  FEWSHOT RECALIBRATION

We have shown that LMs may not be well-calibrated for meaningful slices of broader distributions even if they are calibrated for the broader distribution itself. To tackle the miscalibration problem, we formulate the fewshot recalibration task which takes $k$ unlabeled queries ($x_{1:k}$) drawn from a fine-grained distribution $p_i(x)$ and recalibrates LM predictions with respect to this slice. In practice, the first few queries in a chat session can provide a sketch of a given user's query distribution (e.g. questions about abstract algebra), and thus be used for distribution-specific recalibration in this framework.

To recalibrate predictions, we train a fewshot recalibrator $f_\theta \colon x_{1:k} \to g$ which outputs a function $g$ that adjusts model confidence $g \colon p_{\text{LM}}(\hat{y} \mid x) \to p'_{\text{LM}}(\hat{y} \mid x)$, while the underlying model $p_{\text{LM}}$ and its predictions remain unchanged. Next, we discuss our choice for the target function $g$ and details for training the recalibrator, as depicted in Figure 3.

Figure 3: An illustration of the fewshot recalibrator. This model learns to predict the precision curve for slices (e.g. psychology only, or 20% psychology-80% biology) of a broader distribution (mix of psychology, biology, botany etc.), using fewshot unlabeled examples. At test time, it can predict the precision curve for an unseen slice (e.g. 66% botany-34% biology) given only an unlabeled fewshot set drawn from it. This precision curve can then be used to accomplish various downstream goals.

## 3.1 PREDICTING PRECISION CURVES RATHER THAN CALIBRATION CURVES

The most direct choice for $g$ would be the calibration curve, i.e. a function that adjusts model confidence to predicted accuracy. However, as described in §2.1, calibration curves rely on binning predictions based on confidence estimates. This binning step introduces two hyperparameters: (1) the binning design where scores can be grouped into equally-spaced bins with equal interval ranges, or equally-sized bins with an equal number of examples per bin. And, (2) the number of bins such that scores can be grouped into a large number of bins each containing a small number of examples, or a small number of bins each containing many examples. Both hyperparameters affect the shape of the calibration curve, and certain choices can hide miscalibration issues, making this an unreliable prediction target for the calibrator.

Instead, we propose predicting the precision curve (PC; $\text{prec}(\cdot)$), which maps confidence thresholds to precision scores. So, $\text{prec}(0.5) = 0.8$ means that for all the examples with confidence greater than 0.5, the model $p_{\text{LM}}$ achieves a precision of 0.8. In contrast to the calibration curve, the precision curve has no hyperparameters. It is also extremely flexible. For instance, it can be converted to the corresponding calibration curve with any hyperparameter setting, given additional information about the distribution over confidence scores (see details in §3.3). Conversely, it is hard to convert a calibration curve to a precision curve since the binning step is lossy. This flexibility allows us to accomplish a variety of downstream goals such as reducing calibration error, finding optimal confidence thresholds for desired precision etc. as described in §3.3. For this reason, we choose precision curves as our calibrator's prediction target $g$.

## 3.2 TRAINING THE FEWSHOT RECALIBRATOR

We train a fewshot recalibrator $f_\theta$ that takes a small set of $k$ unlabeled examples from some fine-grained distribution slice $p_i$, and predicts the precision curve for $p_i$. The training set consists of many slices and each example is a pair $(x^i_{j,(1:k)}, \text{prec}^i)$ which is the $j$-th few-shot set drawn from the $i$-th slice mapping to the corresponding ground-truth precision curve. The training loss minimizes $L_2$ distance between the ground-truth and predicted precision values at different confidence thresholds.

While the training loss penalizes all errors equally, over-estimating precision at some confidence threshold can be seen as a more costly error than under-estimating it. This is because predicting a higher precision score than the ground-truth means the recalibrator believes the model correctly answers more questions than it actually can, and the confidence threshold does not trigger abstention when it should. Conversely, when under-estimating precision, the confidence threshold is more conservative and sacrifices recall in favor of more reliable answers. In this work, we prioritize correctness over recall, as is likely in most practical scenarios, by adapting the $L_2$ training objective to be asymmetric.

$$\mathcal{L}(\theta) = \begin{cases} \beta ||\text{prec}_\theta(c) - \text{prec}(c)||^2 & \text{if } \text{prec}_\theta(c) > \text{prec}(c), \\ ||\text{prec}_\theta(c) - \text{prec}(c)||^2 & \text{otherwise.} \end{cases}$$

where $\text{prec}_\theta = f_\theta(x_{1:k})$ is the output of the fewshot recalibrator. This penalizes over-estimation more than under-estimation by setting the coefficient $\beta > 1.0$.

## 3.3 EVALUATION

Our fewshot recalibrator outputs a precision curve which is flexible and can be used to accomplish various downstream goals. We describe three of them here, along with the corresponding metrics that define success.

**Achieving Target Precision**    For a given system, we may want to guarantee a minimum level of precision. The goal, then, is to identify distribution-specific confidence thresholds that achieve that level of precision without sacrificing much recall. In this setting, we can directly use the predicted precision curve $\text{prec}_\theta$ as a lookup table and find the threshold that attains the target precision. We evaluate performance by measuring the success rate of whether the selected threshold achieves the target precision on the ground-truth precision curve.

**Reducing Calibration Error**    Alternatively, the goal can be to reduce the system's calibration error. For this setting, first we map the predicted precision curve $\text{prec}_\theta$ to the corresponding calibration curve, given the confidence scores of the predictions. We do this as follows: let $\text{count}(a)$ denote the number of examples whose confidence exceeds $a$. For bin $B_i$, we have the upper $B_i.r$ and lower $B_i.l$ bounds on the confidence scores. We compute the accuracy for $B_i$: $\text{acc}(B_i) = \frac{\text{prec}_\theta(B_i.l)\text{count}(B_i.l) - \text{prec}_\theta(B_i.r)\text{count}(B_i.r)}{\text{count}(B_i.l) - \text{count}(B_i.r)}$, which along with the confidence $\text{conf}(B_i)$, is sufficient to recover the calibration curve. Once we have the calibration curve, we can apply histogram binning (Zadrozny & Elkan, 2001) to map confidence scores to the corresponding accuracy, minimizing the calibration error. We evaluate performance by measuring ECE.

**Maximizing Utility**    Another downstream goal in practice can be to maximize the utility of a system, which consists of the abstention cost (sacrifices recall) and the error cost (sacrifices precision). Inspired by the rejection learning framework (Cortes et al., 2016; Bartlett & Wegkamp, 2008), we define a cost function that clearly specifies the trade-off: incorrect predictions incur a cost of 1 and abstaining incurs a cost $c \in [0, 1]$, while correct predictions incur no cost. For a fixed value for $c$, the goal is to maximize utility (i.e. negative cost).

Given the predicted precision curve $\text{prec}_\theta$ and the raw confidence scores for predictions, let $\text{count}(t)$ denote the number of examples whose confidence exceeds $t$ and $N$ denote the total number of examples. Then, we estimate the cost at each threshold $t$ as $\text{Cost}(t) = (1 - \text{prec}_\theta(t)) \cdot \text{count}(t) + c \cdot (N - \text{count}(t))$, where the first term accounts for incorrect predictions and the second term accounts for abstentions. And we find the optimal threshold $t^*$ that minimizes $\text{Cost}(t)$ via a grid search over $t \in [0, 1]$. To evaluate the goodness of the selected threshold $t^*$, we assume access to labeled data, and measure the empirical utility achieved by abstaining when the model's confidence is lower than the selected threshold and making a prediction otherwise.

## 4    EXPERIMENTAL SETUP

### 4.1    DATASETS

We evaluate our fewshot recalibrator on two datasets: MMLU (Hendrycks et al., 2021) consists of multiple choice questions categorized into 57 different subjects (e.g. *abstract algebra*, *high school physics*, *law*), each of which serves as a separate domain. XNLI (Conneau et al., 2018) is a natural language inference task, where the model predicts if the given hypothesis entails, contradicts or is neutral to the corresponding premise. Examples are categorized into 10 genres (e.g. *travel guides*, *speeches*, etc.) in 15 languages each, for a total of 150 domains.

Rather than simply looking at individual domains, we simulate a wider variety of slices of the broader distributions via weighted mixtures of domains. We do this by first sampling a number of domains $m$ from a geometric distribution[2], and then randomly selecting $m$ domains from the full set. Then, we sample mixture weights from a Dirichlet distribution and construct the slice by mixing the $m$ domains according to their mixture weights, e.g. $0.25$ math and $0.75$ history. Once we have constructed these slices, we sample $k$ unlabeled examples from each distribution to serve as the fewshot set that provides a sketch of the corresponding slice. For the main experiments we set $k = 20$, and for ablation studies, we consider $k = \{5, 10, 20, 30\}$.

We sample 20K slices for the training set and 2K unseen slices for the test set, ensuring that examples which appear in the test data's fewshot sets are held out from training. We also construct an UNSEEN test set for XNLI, where 10 domains are entirely held out from the training data and are used to construct a separate set of 2K mixtures.

---

[2]We use Geometric$(0.5)$ for MMLU and Geometric$(0.3)$ for XNLI.

## 4.2 MODELS

We train fewshot recalibrators for PaLM2-Large (Anil et al., 2023) and LLaMA-65B (Touvron et al., 2023) on MMLU and only PaLM2-Large, the best performing model, on XNLI. We also include recalibration results for LLaMA-30B in Appendix B. Our recalibrator is a LLaMA-7B model, finetuned for $4K$ steps for MMLU and $2K$ for XNLI, both with a batch size of 16, a learning rate of 2e-5 and a cosine learning rate schedule (see more details in Appendix A). All finetuning experiments use 16 A100-40GB GPUs. Recall from §3.2, our training objective is the asymmetric $L_2$ loss, and we set $\beta = 5$ in all experiments.

## 4.3 BASELINES

We compare our fewshot recalibrator against the following baselines which output precision curves.

SAMPLE AVERAGE is the precision curve of the combined distribution over all the domains based on the queries that appear in the training data. This baseline is not distribution-specific: it uses a single curve for all test set distributions.

DOMAIN AVERAGE involves averaging the precision curves for each domain. Similar to sample averaging, this approach is not distribution-specific.

EMPIRICAL uses the precision curve obtained from only the $k$ fewshot *labeled* queries. Note that this baseline has an unfair advantage over other approaches, including ours, because it assumes access to the labels of the $k$ fewshot queries.

ORACLE is the ground-truth precision curve of the corresponding slice's distribution, and serves as a skyline for the best achievable performance for curve prediction approaches.

In the reducing calibration error setting, we compare our approach to the canonical recalibration method of temperature scaling (Guo et al., 2017). Temperature scaling (TS) uses a held out calibration set to select a temperature, and then applies that temperature to the test data. We compare against two variants of temperature scaling, and they differ in the choice of the calibration set.

TS (FEWSHOT) uses the $k$ fewshot examples with ground-truth labels as the calibration set. We run grid search on values for the temperature in $\{0.1, 0.2, \cdots, 1.9, 2.0, 3.0, 4.0, 5.0\}$ to find one that minimizes ECE for the $k$ examples.

TS (ALL DOMAINS) uses the training data, combining all domains, as the calibration set. Similarly, we run grid search on values for the temperature to minimize ECE for the entire training set.

Lastly, in the utility maximization setting, we compare against a baseline inspired by the rejection learning framework.

ABSTAIN finetunes LLaMA-7B to predict the correctness of a classifier $p_{\text{LM}}$. When the abstain model predicts the model will be incorrect, we abstain from answering the question, and otherwise, the question is answered by the classifier $p_{\text{LM}}$.

# 5 RESULTS

## 5.1 ACHIEVING TARGET PRECISION

We first experiment with measuring the success rate of selecting a confidence threshold that achieves a given target precision on the slice's ground-truth precision curve. As shown in Table 1, our fewshot recalibrator outperforms baselines by achieving a higher success rate for three different target precision values of $0.85$, $0.9$ and $0.95$.

In spite of the fact that the Empirical baseline has access to the fewshot example labels, our recalibrator consistently outperforms it by a large margin. This shows that while the fewshot set itself is not sufficient for plotting a precision curve and selecting a slice-specific threshold, our recalibrator successfully learns to infer the full slice's distribution, and its corresponding precision curve, from this this set. This is also demonstrated in Figure 5, where we show examples of precision curves generated by our fewshot recalibrator. As we can see, the Empirical curve deviates far from the Oracle curve, while our recalibrator closely approximates it, and tends to upper bound it, as a consequence of our asymmetric training objective.

| | Target Precision | 0.85 | | 0.9 | | 0.95 | | |
|---|---|---|---|---|---|---|---|---|
| | | **Success** | **Recall** | **Success** | **Recall** | **Success** | **Recall** | $L_2$ |
| XNLI PaLM2-L | **Sample Avg** | 0.47 | 0.86 | 0.55 | 0.71 | 0.62 | 0.42 | 0.001 |
| | **Domain Avg** | 0.53 | 0.86 | 0.55 | 0.71 | 0.62 | 0.42 | 0.001 |
| | **Empirical** | 0.47 | 0.81 | 0.38 | 0.68 | 0.34 | 0.52 | 0.008 |
| | **FSC(Ours)** | **0.69** | 0.83 | **0.75** | 0.66 | **0.76** | 0.37 | **0.001** |
| | Oracle | 1.00 | 0.85 | 1.00 | 0.7 | 1.00 | 0.45 | 0.000 |
| MMLU PaLM2-L | **Sample Avg** | 0.64 | 0.95 | 0.64 | 0.88 | 0.60 | 0.75 | 0.006 |
| | **Domain Avg** | 0.71 | 0.93 | 0.78 | 0.84 | 0.78 | 0.69 | 0.007 |
| | **Empirical** | 0.61 | 0.91 | 0.47 | 0.86 | 0.34 | 0.74 | 0.007 |
| | **FSC(Ours)** | **0.87** | 0.87 | **0.85** | 0.80 | **0.77** | 0.67 | **0.002** |
| | Oracle | 1.00 | 0.91 | 1.00 | 0.85 | 1.00 | 0.74 | 0.000 |
| MMLU LLaMA-65B | **Sample Avg** | 0.58 | 0.60 | 0.59 | 0.51 | 0.57 | 0.36 | 0.012 |
| | **Domain Avg** | 0.72 | 0.57 | 0.80 | 0.41 | **0.99** | 0.02 | 0.012 |
| | **Empirical** | 0.43 | 0.58 | 0.40 | 0.48 | 0.34 | 0.40 | 0.023 |
| | **FSC(Ours)** | **0.90** | 0.50 | **0.89** | 0.39 | 0.80 | 0.23 | **0.006** |
| | Oracle | 1.00 | 0.60 | 1.00 | 0.51 | 1.00 | 0.39 | 0.000 |

Table 1: Our fewshot recalibrator has a higher success rate for identifying confidence thresholds that achieve a given target precision, as compared to the baselines, while maintaining reasonable recall.

Our approach also outperforms the Sample and Domain averaging baselines in all settings but one: for a target precision of 0.95 when calibrating LLaMA-65B on MMLU. However, in this case Domain averaging achieves a high success rate of 0.99 by selecting an extremely high threshold and entirely sacrificing recall, down to 0.02. In contrast, our recalibrator strikes a better balance between achieving the target precision with a high success rate, while still maintaining reasonable recall.

## 5.2 REDUCING CALIBRATION ERROR

| | XNLI (PaLM2-Large) | | | MMLU (PaLM2-Large) | | | MMLU (LLaMA-65B) | | |
|---|---|---|---|---|---|---|---|---|---|
| | **ECE** | **Win%** | **Lose%** | **ECE** | **Win%** | **Lose%** | **ECE** | **Win%** | **Lose%** |
| **Base** | 0.059 | 22 | 78 | 0.063 | 38 | 62 | 0.109 | 16 | 84 |
| **Sample Avg** | 0.049 | 39 | 61 | 0.082 | 17 | 83 | 0.103 | 25 | 75 |
| **Domain Avg** | 0.049 | 39 | 61 | 0.085 | 17 | 83 | 0.107 | 22 | 78 |
| **Empirical** | 0.094 | 9 | 91 | 0.078 | 29 | 71 | 0.122 | 14 | 86 |
| **TS (fewshot)** | 0.094 | 8 | 92 | 0.079 | 27 | 73 | 0.120 | 16 | 84 |
| **TS (all domains)** | 0.057 | 23 | 77 | 0.063 | 38 | 62 | 0.099 | 24 | 76 |
| **FSC(ours)** | **0.045** | - | - | **0.053** | - | - | **0.074** | - | - |
| Oracle | 0.011 | 99 | 1 | 0.009 | 100 | 0 | 0.016 | 100 | 0 |

Table 2: Our approach achieves the lowest calibration error (ECE), outperforming all baselines. Pairwise comparisons show that it has a lower ECE for most of the test slices, indicated by each baseline's lose percentage. **Base** refers to the LM without any temperature scaling.

For the goal of reducing calibration error, we similarly find that our fewshot recalibrator outperforms baselines by achieving the lowest ECE score across various settings, as shown in Table 2. We also conduct a pairwise comparison and find that our recalibrator wins by achieving a lower ECE score most of the test slices as compared to all other approaches.

We find that the labeled fewshot set is not a useful proxy for the whole slice, since selecting a temperature based on this set for temperature scaling fails to improve ECE over the base LM with a temperature of 1. We also find that selecting a single temperature for all slices, based on the broader distribution of the training set examples, is sub-optimal. In contrast, our fewshot recalibrator can provide slice-specific calibration which results in lower ECE.

## 5.3 MAXIMIZING UTILITY

For the utility maximization setting, we experiment with two values of the abstention costs, $c = 0.4$ which favors abstaining more (i.e. precision) and $c = 0.6$ which favors answering more (i.e. recall). These two settings evaluate each method's flexibility to balance different trade-offs between

|  | XNLI (PaLM2-Large) | | MMLU (PaLM2-Large) | | MMLU (LLaMA-65B) | |
|---|---|---|---|---|---|---|
|  | $c = 0.4$ | $c = 0.6$ | $c = 0.4$ | $c = 0.6$ | $c = 0.4$ | $c = 0.6$ |
| **Abstain** | -0.224 | -0.240 | -0.162 | **-0.188** | -0.315 | -0.390 |
| **Sample Avg** | -0.206 | -0.219 | -0.169 | -0.197 | -0.289 | -0.382 |
| **Domain Avg** | -0.206 | -0.219 | -0.171 | -0.197 | -0.289 | -0.388 |
| **Empirical** | -0.208 | -0.225 | -0.164 | -0.190 | -0.293 | **-0.372** |
| **FSC(Ours)** | **-0.202** | **-0.218** | **-0.157** | -0.189 | **-0.284** | **-0.372** |
| Oracle | -0.192 | -0.213 | -0.150 | -0.180 | -0.277 | -0.358 |

Table 3: Our fewshot recalibrator is better at maximizing utility, and thus, finding the right balance between abstaining and making predictions.

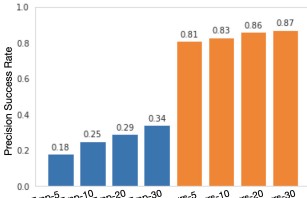 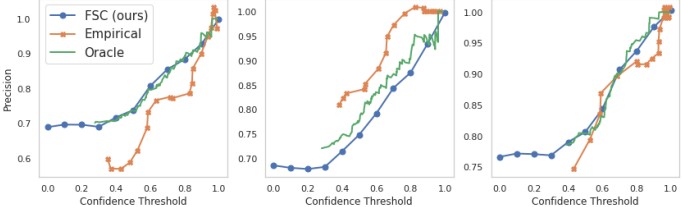

Figure 4: Our approach works well even with small fewshot sets.

Figure 5: Examples of precision curves generated by the fewshot recalibrator, compared to the Empirical and Oracle curves. Our curves approximate the Oracle curves more closely.

precision and recall. As shown in Table 3, we find that our fewshot calibrator strikes a good trade-off between precision and recall for both settings, consistently achieving a higher utility as compared to baselines, including the Abstain model.

## 5.4 Extrapolation to Unseen Domains

| **Target Precision** | **0.85** | | **0.9** | | **0.95** | | |
|---|---|---|---|---|---|---|---|
|  | **Success** | **Recall** | **Success** | **Recall** | **Success** | **Recall** | $L_2$ |
| **Sample Avg** | 0.60 | 0.86 | 0.63 | 0.70 | 0.38 | 0.42 | 0.002 |
| **Domain Avg** | 0.65 | 0.85 | 0.63 | 0.70 | 0.38 | 0.42 | 0.002 |
| **Empirical** | 0.53 | 0.81 | 0.43 | 0.69 | 0.33 | 0.53 | 0.009 |
| **FSC(Ours)** | **0.79** | 0.83 | **0.74** | 0.67 | **0.69** | 0.34 | 0.001 |
| Oracle | 1.00 | 0.87 | 1.00 | 0.72 | 1.00 | 0.43 | 0.000 |

Table 4: Precision Success Rate On Unseen Domains from XNLI. Our approach achieves the highest success rate and lowest $L_2$ distance on previously unseen domains, without sacrificing much recall.

We also evaluate the extrapolation performance of our fewshot recalibrator. For this, we measure the success rate of achieving target precision on domains from XNLI that were *unseen* in the training set. Table 4 shows that our approach performs well on unseen domains as well, achieving the highest success rate of all curve prediction baselines, while maintaining a reasonable recall.

## 6 Ablation Studies

We run all ablation experiments on the MMLU dataset, recalibrating the PaLM2-Large model.

**Number of fewshot examples** We examine the impact of the number of fewshot examples by experimenting with $k = \{5, 10, 20, 30\}$. As shown in Figure 4, the success rate of achieving target precision increases as we increase the number of fewshot examples for both the Empirical baseline and our fewshot recalibrator,. Our approach with only 5 examples still achieves a high success rate of 0.81, suggesting that our approach is highly suitable for settings with very small amounts of recalibration data.

**Asymmetric vs. symmetric loss** The asymmetric objective penalizes over-estimation of precision more severely than under-estimation. In this ablation experiment, we verify the effectiveness for the asymmetric objective. We find that training our recalibrator with the asymmetric loss ($\beta = 5$) results

in a higher success rate of $0.85$ whereas the symmetric loss only achieves $0.68$, when aiming for a target precision of 90%.

**Performance for different numbers of domains per slice**  Our experiments involve constructing slices using different numbers of domains. Here, we decompose target precision success rate results for mixtures containing 2, 3, 4 and 5 domains. Table 5 shows that performance does not vary significantly across these settings.

|  | 2 domains | | 3 domains | | 4 domains | | 5 domains | |
|---|---|---|---|---|---|---|---|---|
|  | Success | Recall | Success | Recall | Success | Recall | Success | Recall |
| **Empirical** | 0.39 | 0.68 | 0.40 | 0.65 | 0.34 | 0.71 | 0.29 | 0.70 |
| **FSC(ours)** | 0.76 | 0.66 | 0.75 | 0.65 | 0.77 | 0.65 | 0.71 | 0.66 |
| Oracle | 1 | 0.70 | 1 | 0.69 | 1 | 0.71 | 1 | 0.70 |

Table 5: Model performance is robust to the number of domains included in the slice and the success rate does not vary significantly as the number of domains changes.

# 7 RELATED WORK

We note that our fewshot recalibrator draws inspiration from Lee et al. (2021) who introduced this type of meta-learning on slices for the purposes of synthesizing new examples. Below, we discuss relevant prior work on calibration for LMs and abstention.

**Calibration for LMs**  Calibration ensures the model's confidence reflects the model's accuracy, which is instrumental for understanding when to trust LMs. Pretrained language models appear mostly well-calibrated on broader distributions (Kadavath et al., 2022; Xiao et al., 2022; Kuhn et al., 2023), and can express their uncertainty in words (Lin et al., 2022; Mielke et al., 2022; Tian et al., 2023; Zhou et al., 2023). However, the models are still miscalibrated in some settings (Wang et al., 2020; Stengel-Eskin & Durme, 2023), and prior work has focused on recalibrating neural networks by temperature scaling (Guo et al., 2017), Platt scaling (Platt, 1999), isotonic regression (Niculescu-Mizil & Caruana, 2005; Zadrozny & Elkan, 2002), or histogram binning (Kumar et al., 2019; Zadrozny & Elkan, 2001). In this work, we identify a class of miscalibration problems on narrower distributions covering only a few domains, and propose a new framework of distribution-specific recalibration that relies on fewshot, unlabeled queries.

**Abstention**  When the model is not confident about an answer, abstention or deferral to an expert are desirable alternatives compared to responding with the incorrect answer. In order to decide when to abstain, the line of work called rejection learning (or selective classification) focuses on *jointly* learning a rejection function and a predictor (Tortorella, 2000; Santos-Pereira & Pires, 2005; Bartlett & Wegkamp, 2008; Cortes et al., 2016; Geifman & El-Yaniv, 2017; Fisch et al., 2022). The rejection function decides when to abstain, and if the rejection function decides not to abstain, the predictor answers the question. In this paper, we freeze the base LM which functions as the predictor because it is computationally expensive to update a large model for downstream tasks. Instead, we make the abstention decision using a smaller model and the raw confidence of the base LM. Specifically, we use the trained recalibrator to derive a confidence threshold of abstention. To compare to the rejection learning line of work, we train an abstention baseline by learning a rejection function that predicts the correctness of the base model's responses. This resembles the setting of one-stage version of Trapeznikov & Saligrama (2013), which also freezes the base classifier model.

# 8 CONCLUSION AND FUTURE WORK

We have shown that while LMs appear to be well-calibrated on broad distributions, they remain miscalibrated for meaningful slices of that broader distribution. To recalibrate them for each slice, we propose fewshot recalibration which takes fewshot, unlabeled queries and predicts a slice-specific precision curve. We then use the predicted precision curve for three downstream calibration tasks, finding that our approach consistently outperforms existing recalibration methods under all evaluation settings. Future work should study fewshot recalibration for natural language generation tasks, to steer model generated text to be more or less conservative, as well as apply this approach to a broader set of models, including instruction-tuned and RLHF models, and multimodal settings.

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

## A    HYPERPARAMETERS

For inference of LLaMA-65B and LLaMA-30B to obtain the target precision curves, we use the deepspeed library (Rasley et al., 2020) with 4 A-100 GPUs. For training the fewshot recalibrator, we finetune LLaMA-7B using the AdamW optimizer and a cosine learning rate schedule. We use a warmup ratio of 0.03, learning rate of $2e - 5$, and batch size of 16. We train for 4K steps for the MMLU experiments and 2K steps for the XNLI experiments. Our fine-tuning is conducted on 16 A100 GPUs of 40GB memory, and we use Deepspeed Stage 3 to ensure the 7B model fits on GPU. Our implementation of inference and finetuning are based on the Hugging Face library (Wolf et al., 2019).

## B    ADDITIONAL RESULTS (LLAMA-30B)

In addition to LLaMA-65B and PaLM2-Large, we also apply our fewshot recalibrator approach to LLaMA-30B to study the impact of model scales. See results in Table 6, Table 7, and Table 8. Compared to other base models (LLaMA-65B model and PaLM2-Large), we observe similar trends in the minimizing ECE and maximizing utility experiment: We find that our approach outperform all baselines in achieving the lowest calibration error with the highest win rate (Table 7). In addition, our approach outperform all baselines in selecting an abstention threshold that yields the highest utility score (Table 8). The only exception happens for the precision success rate experiment. Unlike the results of LLaMA-65B where our fewshot recalibrator outperform all the baselines including Domain Avg, for LLaMA-30B, Domain Avg achieves higher success rate than our fewshot recalibrator. The gap is particularly large for a target precision of 0.95. We hypothesis that this is because the LLaMA-30B suffers from lower accuracy compared to larger models. Thus, in the training data, the groundtruth precision curve of many custom distributions fail to hit the 95% precision level, leading to a sparsity of training data that hits the 95% precision level. As a result, when we try to infer about 95% precision level at inference time, the model predictions are more prone to error.

| | Target Precision | 0.85 | | 0.9 | | 0.95 | | |
| | | Success | Recall | Success | Recall | Success | Recall | $L_2$ |
|---|---|---|---|---|---|---|---|---|
| MMLU LLaMA-30B | Sample Avg | 0.57 | 0.45 | 0.58 | 0.36 | 0.59 | 0.26 | 0.012 |
| | Domain Avg | 0.76 | 0.38 | 0.72 | 0.32 | 0.94 | 0.09 | 0.013 |
| | Empirical | 0.36 | 0.5 | 0.34 | 0.42 | 0.28 | 0.35 | 0.030 |
| | FSC (ours) | 0.75 | 0.35 | 0.68 | 0.26 | 0.52 | 0.16 | 0.007 |
| | Oracle | 1 | 0.46 | 1 | 0.38 | 1 | 0.28 | 0 |

Table 6: Precision Success Rate for LLaMA-30B on MMLU. Domain Avg achieves higher success rate than our fewshot recalibrator. The gap is particularly large for a target precision of 0.95. We hypothesizes that this is because the LLaMA-30B suffers from lower accuracy compared to larger models (LLaMA-65B). Thus, in the training data, the groundtruth precision curve of many custom distributions fail to hit the 95% precision level, leading to a sparsity of training data that hits the 95% precision level. As a result, when we try to infer about 95% precision level at inference time, the model predictions are more prone to error.

| Method | ECE | win% | lose% |
|---|---|---|---|
| Base | 0.093 | 0.2425 | 0.7575 |
| Sample Avg | 0.106 | 0.2325 | 0.7675 |
| Domain Avg | 0.109 | 0.192 | 0.808 |
| Empirical | 0.131 | 0.091 | 0.909 |
| TS (Fewshot) | 0.117 | 0.187 | 0.813 |
| TS (all domains) | 0.090 | 0.283 | 0.717 |
| FSC(ours) | 0.074 | - | - |
| Oracle | 0.016 | 0.9975 | 0.0025 |

Table 7: ECE for LLaMA-30B on MMLU. Our approach outperforms all the baselines in achieving the lowest calibration error with the highest win rate.

| | | $c = 0.4$ | | | | $c = 0.6$ | | | |
| | | Utility | Win | Tie | Lose | Utility | Win | Tie | Lose |
|---|---|---|---|---|---|---|---|---|---|
| XNLI PaLM2-L | Abstain | -0.352 | 0.3065 | 0.001 | 0.6925 | -0.437 | 0.4595 | 0.002 | 0.5385 |
| | Sample Avg | -0.326 | 0.231 | 0.212 | 0.557 | -0.443 | 0.2445 | 0.1345 | 0.621 |
| | Domain Avg | -0.329 | 0.185 | 0.145 | 0.67 | -0.451 | 0.1985 | 0.0905 | 0.711 |
| | Empirical | -0.329 | 0.279 | 0.0805 | 0.6405 | -0.431 | 0.4105 | 0.1065 | 0.483 |
| | FSC(ours) | -0.319 | 0 | 1 | 0 | -0.428 | 0 | 1 | 0 |
| | Oracle | -0.311 | 0.8125 | 0.13 | 0.0575 | -0.416 | 0.8215 | 0.099 | 0.0795 |

Table 8: Utility Scores for LLaMA-30B on MMLU. Our approach outperforms all baselines in selecting abstention thresholds that yield the highest utility scores.

## C    ADDITIONAL RESULTS (MAXIMIZING UTILITY)

Recall in §5.3, we report the utility score for 3 different settings (LLaMA-65B on MMLU, PaLM2-L on MMLU, and PaLM2-L on XNLI). Here, we provide additional pairwise comparison results that contains win/tie/lose rate of each baseline v.s. our approach in Table 9.

| | | $c = 0.4$ | | | | $c = 0.6$ | | | |
| | | Utility | Win | Tie | Lose | Utility | Win | Tie | Lose |
|---|---|---|---|---|---|---|---|---|---|
| XNLI PaLM2-L | **Abstain** | -0.224 | 0.4 | 0.0005 | 0.5995 | -0.24 | 0.398 | 0.0035 | 0.5985 |
| | **Curve agg** | -0.206 | 0.183 | 0.3795 | 0.4375 | -0.219 | 0.218 | 0.4975 | 0.2845 |
| | **Fewshot** | -0.208 | 0.332 | 0.0775 | 0.5905 | -0.225 | 0.299 | 0.246 | 0.455 |
| | **FSC(Ours)** | -0.202 | 0 | 1 | 0 | -0.218 | 0 | 1 | 0 |
| | **Oracle** | -0.192 | 0.851 | 0.098 | 0.051 | -0.213 | 0.709 | 0.22 | 0.071 |
| MMLU PaLM2-L | **Abstain** | -0.162 | 0.484 | 0.0015 | 0.5145 | -0.188 | 0.5085 | 0.0015 | 0.49 |
| | **Curve_agg** | -0.171 | 0.188 | 0.2005 | 0.6115 | -0.197 | 0.176 | 0.2355 | 0.5885 |
| | **Fewshot** | -0.164 | 0.3095 | 0.0885 | 0.602 | -0.19 | 0.4205 | 0.0885 | 0.491 |
| | **FSC(Ours)** | -0.157 | 0 | 1 | 0 | -0.189 | 0 | 1 | 0 |
| | **Oracle** | -0.15 | 0.862 | 0.096 | 0.042 | -0.18 | 0.823 | 0.124 | 0.053 |
| MMLU LLaMA-65B | **Abstain** | -0.315 | 0.322 | 0.001 | 0.677 | -0.39 | 0.401 | 0.002 | 0.597 |
| | **Curve_agg** | -0.289 | 0.2715 | 0.2135 | 0.515 | -0.388 | 0.225 | 0.1245 | 0.6505 |
| | **Fewshot** | -0.293 | 0.3105 | 0.091 | 0.5985 | -0.372 | 0.448 | 0.1305 | 0.4215 |
| | **FSC(Ours)** | -0.284 | 0 | 1 | 0 | -0.372 | 0 | 1 | 0 |
| | **Oracle** | -0.277 | 0.787 | 0.139 | 0.074 | -0.358 | 0.817 | 0.088 | 0.095 |

Table 9: Additional utility results, including the pairwise comparisons win/tie/lose rate compared to our approach. Overall, our fewshot recalibrator outperforms all baselines in achieving the highest utility scores, and more winning percentages.

## D    ADDITIONAL RESULTS (EXTRAPOLATION)

Recall in §5.4, we show our fewshot recalibrator extrapolates well to unseen domains as demonstrated by the precision success rate experiments. Here, we provide more evidence, demonstrated by the ECE results in Table 10. Same as the trend in the precision experiment, our approach outperforms all the baselines in achieving the lowest calibration error and more winning percentages in pairwise comparison.

| Method | ECE | Win | Lose |
|---|---|---|---|
| **Base** | 0.064 | 0.268 | 0.732 |
| **Sample Avg** | 0.052 | 0.4525 | 0.5475 |
| **Domain Avg** | 0.052 | 0.444 | 0.556 |
| **Empirical** | 0.093 | 0.115 | 0.885 |
| **TS (Fewshot)** | 0.095 | 0.1285 | 0.8715 |
| **TS (all domains)** | 0.061 | 0.3155 | 0.6845 |
| **FSC (ours)** | 0.049 | - | - |
| **Oracle** | 0.011 | 0.9965 | 0.0035 |

Table 10: Unseen ECE Evaluation. Our approach outperforms all the baselines in achieving the lowest calibration error and more winning percentages in pairwise comparison.

