# OpenReview forum: "Calibrated on Average, but not Within Each Slice: Few-shot Calibration for All Slices of a Distribution"
_ICLR.cc/2024/Conference — Submitted to ICLR 2024_

### Official Review · Reviewer_nvLA · 2023-10-30

**Soundness:** 2 fair
**Presentation:** 3 good
**Contribution:** 2 fair
**Rating:** 5
**Confidence:** 3

**Summary:**

The paper presents a technique for calibrating model output confidence, with respect to the inference domain, through few-shot learning of a recalibrator function. In particular, the goal of the paper is to imbue a degree of uncertainty with respect to any underexposed domains. Empirical results showcase improved precision and calibration error for the recalibrated model.

**Strengths:**

1. The paper is well structured and presents a clear motivation for the need of a recalibrator.
2. Experimental results showcase the effectiveness of the recalibrator for improving precision and calibration error.
3. Additional ablation studies further explore the inner workings of the recalibrator.

**Weaknesses:**

1. A few portions of the paper remain unclear to the reviewer (see specific question below).
2. The lack of prior works for comparison makes less clear the effectiveness of the recalibrator.

**Questions:**

The reviewer would like some clarification with regard to the specifics of the recalibrator. In particular, what does the learned recalibrator parameters look like? Does the recalibrator learn just a temperature scaling term?

---

> ### Author Response · Authors · 2023-11-23
>
> We thank reviewer nvLa for their helpful review. We will incorporate their suggestions in the next version of this paper and clarify the main comments below.
>
> **The lack of prior works for comparison makes less clear the effectiveness of the recalibrator.**
> We thank the reviewer for this suggestion. We will include more discussions and comparisons with prior works in the next version of the paper.
>
> **"Does the recalibrator learn just a temperature scaling term?"**
> No. From a high level, our recalibrator learns to predict the precision curve for a specific domain of questions (e.g., math questions, or history questions). We then use the precision curve to recalibrate the model’s raw confidence and choose when to abstain.
>
> **What does the learned recalibrator parameters look like?**
> We finetune a LLaMA-7B model to serve as the recalibrator even for larger LMs. So, the recalibrator model parameters are the post-finetuning weights of LLaMA-7B.

---

### Official Review · Reviewer_fySy · 2023-10-31

**Soundness:** 2 fair
**Presentation:** 2 fair
**Contribution:** 2 fair
**Rating:** 5
**Confidence:** 3

**Summary:**

The paper addresses the issue of calibration in language models (LMs), where a model's confidence in its predictions accurately reflects the likelihood of correctness. It highlights that while LMs may be well-calibrated on average, they can be miscalibrated within narrower subsets of the distribution. The paper introduces a framework for calibrating models on specific slices of the distribution using only a few unlabeled samples. The proposed fewshot recalibrator consistently outperforms existing calibration methods in various evaluation settings.

**Strengths:**

1) The paper introduces an innovative approach for improving LM calibration, especially within narrower slices of the data distribution, which is essential for specific user queries or applications.
2) The paper demonstrates that the fewshot recalibrator consistently outperforms existing recalibration methods across various evaluation settings, showing its reliability and effectiveness.
3) The provided approach is simple and easy to follow.

**Weaknesses:**

The results in this paper lack theoretical support, raising me some questions (see Questions).

**Questions:**

1) The proposed method is simple to follow. What is the limitation of this method? It is important to see the boundaries of method use.
2) I wonder if the problem of this paper would vanish as the data and model become larger. Could you please demonstrate it?

I will consider raise the score if author can address my question.

**Details Of Ethics Concerns:**

nan

---

> ### Author Response · Authors · 2023-11-23
>
> We thank reviewer fySy for their helpful review. We clarify their major questions below:
>
> **What is the limitation of this method?**
> Like many few-shot learning methods, our method can struggle when given target domains that are outside of the distribution seen during training. Our recalibrator is trained via meta-learning framing to generalize to different domains of MMLU questions (e.g., from math questions to history questions). Given this training data, our recalibrator could fail to generalize to more distant distributions (e.g., beyond academic question such as cooking questions).
>
> **I wonder if the problem of this paper would vanish as the data and model become larger. Could you please demonstrate it?**
> The experiments in the paper already include large models, e.g., PaLM2-Large and LLaMA-65B. In our experiments, the calibration problem continues to exist even at these larger model scales. We will discuss the impact of scaling, as larger LMs are easier to recalibrate with few-shot examples, in the next version of the paper.

---

### Official Review · Reviewer_LrPN · 2023-11-03

**Soundness:** 2 fair
**Presentation:** 3 good
**Contribution:** 3 good
**Rating:** 6
**Confidence:** 4

**Summary:**

The paper discusses a novel approach for calibrating the confidence estimates of language models (LMs). While recent research has made progress in extracting well-calibrated confidence estimates from LMs, the authors highlight that LMs can still exhibit significant miscalibration within specific subsets of data. For instance, a model may appear well-calibrated overall but can be overconfident in one subject and under-confident in another. This miscalibration is particularly problematic because users often focus on narrower slices of the data distribution.
To address this issue, the authors propose a new framework for calibrating LMs on specific slices of the data distribution using only a few unlabeled samples from that slice. They train a model to approximate the precision-threshold curve for a given slice by using these few-shot samples to predict the LM's empirical precision at various confidence thresholds. This approach allows them to identify slice-specific confidence thresholds above which the LM's predictions can be trusted and below which it should abstain. Additionally, they demonstrate that the precision curve can be linked to the classic calibration curve, offering guidance on adjusting the LM's confidence to achieve lower calibration error.
Experimental results show that their few-shot recalibration method consistently outperforms existing calibration methods. For example, it improves the calibration error for a specific LM, PaLM2-Large, on a particular dataset (MMLU) by 16% compared to temperature scaling. This approach helps enhance the reliability of LM predictions within narrower data distributions, which can be valuable in various applications.

**Strengths:**

1. The problem is well motivated. Figures 1, 2 and discussion with examples of domains is helpful in understanding the slices of distributions and the miscalibration problem within slices.

2. The paper provides a novel method based on precision-curves to recalibrate the model using few unlabeled samples from the domain/slice. The proposed method is few-shot, it does not require any additional labeled data and retraining to recalibrate the model.

3. These examples are used to train a separate recalibrator $f_\theta$ that learns to predict the precision curve for a given slice.
Empirical evaluation on MMLU and XNLI benchmarks is provided, showing that the proposed method is achieving better calibration per slice in comparison to the common baselines.

**Weaknesses:**

Weaknesses and questions,

1. While the first 2 sections are written well, I found section 3 lacking some details and precision. Could you have a more elaborate section 3 with the following,

    a.  Provide technical details of what is meant by a slice and how they are constructed for training $f_\theta$.

    b.  Provide an algorithm block listing all the steps involved in training (starting from how the data slices are constructed) to outputting the recalibrated scores.

    c. Could you give a technical definition of the precision curve, it is there but in loose words. How is this curve estimated for each slice? I do get it roughly from the paper but would be better if a precise mathematical expression/definition is provided.

    d.  In 3.2 what is the loss computed on? It only has $\theta$ (parameters) but data is missing.


2. It seems the method uses some set of precision curves $p_i$ (corresponding to some slices/domains) and tries to predict the precision curve for some given slice. So if those curves $p_i$ are not good, the method would not be able to correct them, since it is just using some unlabeled data points to identify the domain or the relevant $p_i$ to combine in some way.


3. Could you provide some simulations (on synthetic data) demonstrating the precision curves for slices and how the proposed method actually does recalibration.

4. Are the experiments not having any randomness? If not, could you please provide variance or IQR etc. for the numbers reported?

**Questions:**

Please see the weaknesses section above. I am also curious to know if the authors tried some other alternatives for $\beta$ in the loss function. Instead of $\beta$ why not use $(prec_{\theta}(c) - prec(c))$ or some other increasing function of this difference. I mean

$ u(\theta,c) = \exp( (prec_{\theta}(c) - prec(c)) )$ and use it in both terms of the loss function as follows,

$\mathcal{L}(\theta) = u(\theta, c) ||(prec_{\theta}(c) - prec(c)) ||_2^2$

Not sure if this is in the right direction, but at a high level I am trying to see if you can get rid of the "if" condition from the loss and make it a smooth function.

---

> ### Author Response · Authors · 2023-11-23
>
> We thank LrPN for the review and helpful suggestions. We will incorporate them in the next version of the paper and respond to major comments below:
>
> **Rewrite Section 3 (Method section) to be more elaborate.**
> We agree that the method section lacks clarity and will rewrite the section with greater technical detail about both the learning settings and our methods, following LrPN’s suggestions.
>
> **So if those precision curves are not good, the method would not be able to correct them**
> We wish to clarify a misunderstanding: the goal of our method is not to improve a model’s precision curve, but rather to predict it, and use the predicted curve to choose when to abstain: From a high-level, models with “bad” precision curves, meaning that they are low accuracy, should abstain more, and models with “good” precision curves should abstain less. Intuitively, we predict which regime we are in and abstain appropriately.
>
> **"Could you provide some simulations (on synthetic data) demonstrating the precision curves for slices and how the proposed method actually does recalibration."**
> We will rewrite the method section to clarify this and specifically provide details of the recalibration procedure in the next version of the paper.

---

### Official Review · Reviewer_QbDV · 2023-11-03

**Soundness:** 3 good
**Presentation:** 1 poor
**Contribution:** 1 poor
**Rating:** 3
**Confidence:** 4

**Summary:**

The paper demonstrates that LMs, while calibrated across domains, could be miscalibrated on individual domains. A method is proposed to recalibrate separately on individual domains using a few labeled samples from the individual domain. It is shown that the method perform well compared to baselines.

**Strengths:**

The problem is well-motivated and the phenomenon of "calibrated on average but not within a slice" is demonstrated well. For readers outside the calibration community, this is a nice demonstration of the fact that some sort of "averaging" is necessary to achieve calibration.

The proposed solution is consistent with findings in the calibration literature.

**Weaknesses:**

In my opinion, the paper is an interesting experimental work. However, it lacks the methodological novelty, expositional quality, or experimental rigor required for academic publication.

## Novelty
The observation that classifiers can be "calibrated on average but not within a slice" is not novel. It is well-understood in the community that calibrating for a given target could lead to miscalibration for targets at finer granularity. This is true even within the same domain when looking at different classes, leading to unexpected behavior of the kind demonstrated in Sec 2. For one exposition, see this paper: https://arxiv.org/abs/2107.08353. A standard solution is then to recalibrate for the finer-granularity targets using a small supervised set adapted to that target, as the paper does.
Using the precision curve instead of the reliability curve is also not new; something similar was done in this paper: https://arxiv.org/abs/2006.12800.
Overall, the concerns and proposed resolution are consistent with, but not very different from, existing research in the calibration space.

## Experiments
Although the change in metric to precision is somewhat well-motivated, it makes things murky since the improved performance could be because the method is tailored to that metric. Since the paper is written as a calibration paper, and not an improve precision@k paper, I encourage to report standard metrics like accuracy and classwise-ECE (apart from conf-ECE) in order to have a fair comparison.
The benchmarking to previous methods is severely lacking. The only other calibration method compared to is temperature scaling (TS), which is far from the state-of-the-art. There are tons of calibration methods, and I don't want to recommend a specific one since it is not clear which are the "best ones" (the authors should do they our own lit survey beyond Guo et al.). Either ways, it is quite clear that TS performs very poorly once we move beyond conf-ECE. Further, the setting of the comparison to TS is not fully clear to me (elaborated in the next question).

## Writing
The final method or even the learning setting is still not clear to me, including basic questions highlighted in the next category. The actual method, as far as I can understand, is contained in Sec 3.2, which can do with much more detail.
In the experiments, what is FSC? After expending some energy, it becomes clear that FSC = Few-Shot-Calibrator, but such oversights show a lack of regard for the readers'/reviewers' time.

**Questions:**

The final method used is still quite unclear to me.
- **Supervised or not?** Based on Sec 3.2, it seems that you require access to the ground-truth labels to compute the true precision curve. However, in Sec 4.3, under "Empirical" you say that access to the labeled queries gives it an unfair advantage. The TS fewshot baseline also uses the labels.
- **How is TS trained?** Is a separate TS temperature learnt for every slice or all the categories merged to learn one temperature? If it's the latter, the underperformance is not surprising at all. Either ways, is not a strong baseline.
- **Section 5.1?** Could you paraphrase the goal of this section? I don't understand "We first experiment with measuring the success rate of selecting a confidence threshold that achieves a given target precision on the slice’s ground-truth precision curve".
- **Accuracy values?** Does the method sacrifice accuracy in favor of ECE and utility as given c's? The improvement in utility shown in Sec 5.3 is quite small, and ECE can be easily improved by sacrificing accuracy or the so-called "sharpness" criteria.
- **Other notions of ECE?** Please consider and report other forms of calibration error such as classwise-ECE, NLL, etc.

---

> ### Author Response · Authors · 2023-11-23
>
> We thank reviewer QbDV for their review and helpful suggestions. We will incorporate these in the next version of our paper, and clarify the reviewer’s questions below.
>
> **A standard solution is then to recalibrate for the finer-granularity targets using a small supervised set adapted to that target, as the paper does**
>
> The novelty of our approach is that we meta-learn the recalibration in a few-shot manner. In the referenced standard solution, the small supervised set still requires a non-trivial number of examples with ground-truth labels in the target domain. In contrast, **our approach removes the need for supervised labels entirely** and can recalibrate with **as few as 5 unsupervised examples**. Specifically, each training example in our approach consists of 5 to 20 unlabeled examples and a set of labeled examples that are used to draw the precision curve.  Then, our approach can accurately predict the precision curve and consequently recalibrate predictions for a new domain, given only 5 to 20 unsupervised examples.
>
> **Consider other forms of calibration errors (e.g., NLL, classwise-ECE, accuracy of answered set)**
> We thank the reviewer for this suggestion, we will incorporate NLL, Brier score, and accuracy in the future version of the paper.
> Since language models typically output only one answer, users mainly care about calibration over the highest probability answer, i.e., top-label calibration. Hence, we focus on improving top-label calibration rather than classwise calibration, such as classwise-ECE.
>
> **The benchmarking to previous methods is severely lacking.**
> We acknowledge that the submission misses important related work, which we will include in the next version of the paper.
>
> **The final method or even the learning setting is still not clear to me**
> We acknowledge important missing details from the method section. We will revise this section with more detail about both the learning setting and our method. We answer the reviewer’s specific questions below:
>
>  - **What supervision do we use in our method?**
> During training, we leverage both labeled and unlabeled examples: we learn a precision curve from a set of labeled examples, and then learn to predict it from a small number of unlabeled examples. This enables us to predict the precision curve in a new domain at inference time from only a small number of unlabeled examples, and no labeled examples.
>
>  - **How is temperature scaling trained?**
> Our temperature scaling baseline TS (fewshot) learns a separate TS temperature for every slice. TS (all domains) learns a temperature with all domains merged.
>
>  - **What is the goal of section 5.1?**
> Our high-level goal is to produce language models that can choose to abstain when asked about information they don’t know, hence avoiding “hallucination” or misinformation. We achieve this in Section 5.1 by finding a threshold of the model’s raw confidence such that the model answers the question if the predicted raw confidence exceeds the threshold and abstains otherwise. We choose the confidence threshold that achieves a certain precision target, while maintaining as high recall as possible.

---

### Author Response · Authors · 2023-11-23

We thank all the reviewers for their time and helpful suggestions. The method section in the submission lacks clarity, which led to significant misunderstandings. We will revise this section with more detail about the learning settings and our method. Additionally, we will revise the paper to include more discussion and comparison with missed related works.

---

### Meta-Review · Area_Chair_B1Ar · 2023-12-07

**Metareview:**

This submission sparked interest from reviewers. The topic was seen as interesting. However, it was not clear that the submission meets the bar for ICLR. Indeed, the contribution is felt not fully solid in its exposition and positioning: a little bit incremental, for instance given that recalibration from the precision curve has already been studied. More benchmarking was mentioned as an area for improvement. Finally, the manuscript had some clarity shortcomings.

**Justification For Why Not Higher Score:**

The manuscript feels not solid enough for ICLR. It is not completely possible to tell if there are good ideas in this work because the arguments, theoretical and empirical, are not polished enough.

**Justification For Why Not Lower Score:**

N/A

---

### Decision · Program_Chairs · 2024-01-16

Reject